# Hepatocellular Carcinoma: Beyond the Border of Advanced Stage Therapy

**DOI:** 10.3390/cancers16112034

**Published:** 2024-05-27

**Authors:** Yusra Zarlashat, Shakil Abbas, Abdul Ghaffar

**Affiliations:** 1Department of Biochemistry, Government College University Faisalabad, Faisalabad 38000, Pakistan; yusrazarlashat@gcuf.edu.pk; 2Gomal Center of Biotechnology and Biochemistry (GCBB), Gomal University, Dera Ismail Khan 29050, Pakistan; drshah@gu.edu.pk

**Keywords:** hepatocellular carcinoma, hepatocytes, immune system, multikinase inhibitors, immune checkpoint inhibitors, immune microenvironment, combination therapy

## Abstract

**Simple Summary:**

Hepatocellular carcinoma is the sixth most common cancer globally and is considered one of the deadliest health issues worldwide. The rapid emergence of HCC as a global health crisis has undergone a revolutionary change in treatment strategies. Food and Drug Administration-approved drugs such as tyrosine kinase inhibitors, immune checkpoint inhibitors, and angiogenic inhibitors, as monotherapy and in combination therapies have shown significant advancement to enhance patient outcomes across HCC stages. The effect of these treatments on cellular signaling pathways and immune responses offers promising potential for prolonged survival rates.

**Abstract:**

Hepatocellular carcinoma (HCC) is the deadliest emergent health issue around the globe. The stronger oncogenic effect, proteins, and weakened immune response are precisely linked with a significant prospect of developing HCC. Several conventional systemic therapies, antiangiogenic therapy, and immunotherapy techniques have significantly improved the outcomes for early-, intermediate-, and advanced-stage HCC patients, giving new hope for effective HCC management and prolonged survival rates. Innovative therapeutic approaches beyond conventional treatments have altered the landscape of managing HCC, particularly focusing on targeted therapies and immunotherapies. The advancement in HCC treatment suggested by the Food and Drug Administration is multidimensional treatment options, including multikinase inhibitors (sorafenib, lenvatinib, regorafenib, ramucirumab, and cabozantinib) and immune checkpoint inhibitors (atezolizumab, pembrolizumab, durvalumab, tremelimumab, ipilimumab, and nivolumab), in monotherapy and in combination therapy to increase life expectancy of HCC patients. This review highlights the efficacy of multikinase inhibitors and immune checkpoint inhibitors in monotherapy and combination therapy through the analysis of phase II, and III clinical trials, targeting the key molecular pathways involved in cellular signaling and immune response for the prospective treatment of advanced and unresectable HCC and discusses the upcoming combinations of immune checkpoint inhibitors-tyrosine kinase inhibitors and immune checkpoint inhibitors-vascular endothelial growth factor inhibitors. Finally, the hidden challenges with pharmacological therapy for HCC, feasible solutions for the future, and implications of possible presumptions to develop drugs for HCC treatment are reported.

## 1. Introduction

Parenchymal hepatocytes constitute 60–70% of total cells in the liver and secrete a considerable amount of proteins that perform an effective role in metabolism, protein synthesis, hormones, and innate immunity [1]. The complex functionality of these proteins and fighting toxins or xenobiotics makes liver cells vulnerable to hepatocellular carcinoma (HCC) [2]. HCC is the sixth most frequent cancer and the deadliest health problem in the world [3]. Cirrhosis and chronic liver disease are the principal risk factors for developing HCC, for which excessive alcohol intake and viral hepatitis are leading risk factors [4].

Chromosomal alterations, hormonal changes, genetic mutations, and changes in signaling pathways play a role in developing HCC [5,6]. The oncogenic mutation or abnormal molecular pathways due to excessive alcohol consumption increases the ratio of HCC about twofold; therefore, alcohol and obesity jointly develop HCC aggressively [7]. Those with abnormal high-fat content in the liver are more susceptible to developing metabolic-associated fatty liver disease (MAFLD), which is a primary cause of HCC [8]. Alcohol and aflatoxins potentially develop tumors due to their carcinogenic activity [9]. The risk of HCC recurrence may increase with metabolic diseases, as well as a group of conditions such as abdominal obesity, atherogenic dyslipidemia, insulin resistance, and hypertension [10].

Infectious agents are another threat to developing HCC through genetic mutations and alteration in gene expression. Hepatitis B virus (HBV) and hepatitis C virus (HCV) are the most significant causes of cirrhosis and fibrosis, which leads to HCC [11]. The stable genome of HBV has the potential to infect hepatocytes and transmit to progeny cells [12]. The DNA of HBV makes the host’s genome unstable through the remodeling of host DNA that alters the chromosomal structure, abnormally expresses the oncogenes, and inactivates cancer-suppressor genes. It changes the liver by activating cancer-associated signaling pathways and interacting with immune cells to promote tumor development [13]. The frequent mutation in the RNA genome of HCV makes it more potent to target liver cells and mask innate, adaptive immunity, and it induces chronic infection in 70% of cases [14].

The choice of treatment for HCC patients depends upon the stage of HCC, size of the tumor, liver performance, immune system, and response of patients toward different therapies. The recent developments in drugs related to the control of genes and stimulating immune response for solid tumors have shown great potential to increase the life expectancy of HCC patients [15]. Multikinase inhibitors (MKIs) (sorafenib, regorafenib, and lenvatinib), tyrosine kinase inhibitors (TKIs) (cabozantinib), and the immune checkpoint blockade (ICB) therapy to target cytotoxic T lymphocyte-associated protein 4 (CTLA-4), programmed cell death-1 and its ligand (PD-1/PD-L1) have opened up possibilities for immunotherapy of HCC [16]. The use of anti-PD-1/PD-L1 antibodies (pembrolizumab, durvalumab, nivolumab, sintilimab, and camrelizumab) and anti-CTLA-4 antibodies (tremelimumab and ipilimumab) as immunotherapy have shown potential for treating HCC patients. TKIs and immune checkpoint inhibitors (ICIs) have dramatically altered the treatment of advanced HCC; however, some studies have reported a concerning rise in the recurrence of HCC. This article presents a brief overview of innovative therapeutic approaches beyond conventional therapies (surgery, liver transplantation (LT), radiation therapy, and chemotherapy) and provides insight into the efficacy of anti-HCC drugs. The available data also assess evidence-based therapies and focus on therapeutic drugs in phase II and III clinical trials for HCC treatment.

## 2. Emerging Therapeutic Options for HCC Therapy

The treatment of patients with HCC has become more challenging due to advancements in surgery and systemic therapy. A dynamic modification of the current staging-based approaches is needed to enable flexible therapy allocation. Specifically, factors other than oncological staging that affect treatment delivery and resource availability, such as patient frailty, critical tumor location, comorbidity, and multiple liver functional parameters, are becoming more and more important in the treatment of HCC [17]. Potential therapies for early-stage HCC are LT and liver resection (LR) [18]. Besides this, percutaneous options for treatment, including microwave ablation (MWA) and radiofrequency ablation (RFA), have also been nearly recommended [19]. Thermal ablation, surgical resection, and LT are thought to be the first-line treatments for early-stage HCC; however, many HCC patients are not suitable candidates for these procedures because of comorbidities, location of the tumor, and underlying liver disease. Under such circumstances, locoregional therapies, including transarterial radioembolization (TARE) or chemoembolization (TACE), are recommended [20]. RFA involves using two electrodes on a single antenna opposite to each other in a tumor by visualizing the image and supplying an electric current to increase the necrosis of tumor cells [5]. MWA involves using an antenna and transmitting microwaves in a tumor to kill tumor cells through heating. The water particles excited to produce most of the heat through ionic polarization generated a small amount of heat [21]. HCC is aggressive and sometimes diagnosed late, and it requires protecting liver function after surgery; therefore, a neoadjuvant strategy is suggested [22]. Neoadjuvant therapy is a treatment administered before the main therapy (surgery or LT) to reduce the tumor size. In a phase II study, neoadjuvant cemiplimab (anti-PD-1) was administered to 21 patients with resectable HCC, who thereafter had surgical resection. Overall, 3 of the 20 patients who had successful resections had a partial response, indicating possible advantages that should be investigated further in more in-depth trials, while 4 of the patients had considerable tumor necrosis [23]. HCC recurrence rates after locoregional therapy are high, which leads to the investigation of neoadjuvant systemic therapy such as dovitinib, which targets VEGFRs. Neoadjuvant dovitinib (TKI) (500 mg once a day for 1 month) followed by locoregional treatment demonstrated a 48% overall response rate in 24 patients with early- and intermediate-stage HCC. Notably, intratumoral blood flow was significantly reduced despite multiple dosage adjustments [24]. A neoadjuvant randomized trial involved nine early-stage HCC patients receiving 3 mg of nivolumab (anti-PD-L1) on the first day after 14 days and 1 mg/kg of ipilimumab (anti-CTLA-4) on day 1 after 1.5 months for 2 years. According to this research, three of the nine patients had favorable results, with enhanced T-cell infiltration, a pathological complete response rate of 33.3%, and a population of CD8^+^ T cells in the tumor microenvironment that produced an antitumor immune response [25].

Adjuvant therapy is provided after the primary treatment to prevent recurrence. In a randomized trial, 80 patients were divided into two groups: one receiving sorafenib (TKI) after treatment with TACE and the other group receiving a placebo. Patients with sorafenib had manageable side effects and slower cancer progression, with a median time of 9.2 months compared to 4.9 months for the placebo group [26]. In a randomized clinical trial, 31 patients with intermediate-stage HCC received 400 mg of sorafenib two times a day after TACE. It had a 9.2-month time to progression (TTP) without experiencing any side effects, in contrast to the control group [26]. In a randomized phase III clinical study, 543 patients with HCC were administered 200 mg of camrelizumab (anti-PD-1) on day 1 after 14 days in addition to 250 mg of rivoceranib (TKI) one time a day or 400 mg of sorafenib two times a day. The results of the combination treatment of camrelizumab with rivoceranib increased PFS and OS more effectively than sorafenib, with median PFS of 5.3 and median OS of 22.1 months (camrelizumab–rivoceranib group) versus 3.7 and 15.2 months (sorafenib group), respectively [27].

Generally, TKIs are associated with adverse events (AEs) such as hypertension, diarrhea, hand-foot skin reaction, and fatigue. AEs are less severe at intermediate-stage HCC compared to the advanced stage [28]. ICIs can lead to immune-related adverse events (irAEs) such as hepatitis, pneumonitis, colitis, and thyroid disorders [29]. Systemic therapy is recommended for the advanced stage where the tumor has spread beyond the liver and is not treated by surgery or LT. Systemic therapies include targeted therapies such as TKIs, ICIs, and antiangiogenics. TKIs and ICIs have shown efficacy in patients but show some AEs at advanced-stage HCC such as gastrointestinal disturbances, hypertension, and liver toxicity [30]. ICIs can induce irAEs in advanced-stage HCC patients. To ensure therapy tolerability, it is essential to monitor for and manage these toxicities. Dose modifications or interruptions may be necessary to manage these toxicities and improve treatment tolerability [31].

### 2.1. Multikinase Inhibitors

#### 2.1.1. Sorafenib

Sorafenib (MKI) blocks receptors for BRAF, RAF1, PDGFR, VEGFR, FLT-3, and c-KIT responsible for cancer angiogenesis and cell multiplication and increases the rate of apoptosis [32] (Figure 1). It has an antiangiogenic activity that impedes the development of blood vessels, thus slowing down the proliferation of the tumor. RAS/RAF/MEK is one of the significant pathways for HCC progression [33]. The upregulation of this signaling pathway on behalf of growth factors and proteins of the hepatitis virus develops angiogenesis and HCC [34]. Sorafenib inhibits the tumor and angiogenesis by downregulating the signaling pathway and blocks the eIF4E translation factor to be phosphorylated, which leads to expressing oncogenes and upregulating apoptotic proteins [35]. Sorafenib is a major targeted drug that showed curative benefits in advanced HCC; however, some patients do not respond well due to the compensatory mechanism of the PI3K/AKT pathway [36]. 

A phase III randomized trial involved 150 advanced HCC patients receiving sorafenib 400 mg orally two times a day and 76 patients constituting the placebo (control). The results showed 6.5 months of median overall survival (mOS) in sorafenib-treated patients and 4.2 months in the placebo; therefore, sorafenib increased the OS rate for HCC patients [37]. HCC patients have heterogeneous tumor effects in terms of diagnosis, progression, and chemoresistance to the drugs. The size and heterogeneity of the tumor restricted the sorafenib applications [38]. A phase II trial for advanced HCC, combining oxaliplatin with sorafenib 400 mg once every 14 days and sorafenib alone, showed a manageable safety profile and higher objective response rates (ORRs) in combination than monotherapy. However, the improvement in progression-free survival (PFS) was moderate, with comparable 4-month PFS rates between sorafenib–oxaplatin and sorafenib alone (64% vs. 61%). While the regimen exhibited potential benefits, no further studies were planned due to the modest effect of PFS [39]. In the next randomized phase III trial, doxorubicin–sorafenib was administered to 180 HCC patients and sorafenib alone to 176 patients. The mOS was 9.3 months, and PFS was 4 months in the combination therapy group, while these values were 9.4 months and 3.7 months in the monotherapy group, hence demonstrating significant improvement in OS or PFS among patients with advanced HCC [40]. Sorafenib is a targeted therapy, used alone or in combinations for advanced HCC, but its efficacy is limited by different factors, including tumor heterogeneity and resistance pathways. This highlights the need for more research to explore alternative or combination treatment approaches that could yield maximum OS and PFS.

#### 2.1.2. Lenvatinib

Lenvatinib is an oral MKI with multiple targets, i.e., FGFR, VEGFR, PDGFR-α, KIT, and RET. Lenvatinib has an effective antiangiogenic activity and inhibits the signaling pathways of FGF and VEGF [41] (Figure 2). Its antitumor effect suppresses hepatocyte proliferation by obstructing the signals of FGF through the inhibition of FRS2 phosphorylation [42]. The oxygen supply is limited to the site of a tumor, resulting in cell death. Lenvatinib binds to the receptor in a Type V binding manner, thus inhibiting kinase activity [43]. Lenvatinib, rather than sorafenib, depleted the rate of angiopoietin 2 (Ang 2), a protein responsible for increasing the size of the tumor by regulating the TIE2 receptor [44]. The overexpression of VEGF is interrelated to advanced HCC and angiogenesis. The signaling pathway of FGF activates malignancy in the HCC accordingly, and thus FGF is another molecular factor for tumor growth [45]. Lenvatinib also has antiproliferative activity, targeting the FGF pathway in cancer cells [42]. It inhibits angiogenesis more effectively than sorafenib, which might not block the FGF pathway. This effect is assumed to be associated with lenvatinib, which hindered both FGFR and VEGFR [46]. 

A meta-analysis involving 1481 HCC patients aimed to compare the first-line efficacy of sorafenib with lenvatinib. The median OS and median PFS for those on lenvatinib were 13.4 months and 5.88 months, whereas for those on sorafenib, these values were 11.4 months and 4.17 months. Lenvatinib was found to significantly increase the rate of objective response and disease control. There was no significant difference in the OS rate, but PFS significantly improved in lenvatinib-receiving patients. Lenvatinib was shown to significantly increase the rates of objective response and disease control [47]. This phase Ib/II study showed that in patients with advanced HCC, cadonilimab-–lenvatinib had favorable safety profiles along with promising activity, with ORR of 35.5% and PFS varying from 8.6 to 9.8 months [48]. A phase III trial was carried out comparing lenvatinib combined with TACE in 170 patients versus lenvatinib alone in 168 advanced HCC patients. The combined therapy showed significantly longer mOS (17.8 vs. 11.5 months) and PFS (10.6 vs. 6.4 months). The Lenvatinib–TACE group exhibited a higher ORR (54.1% vs. 25%). These findings suggest that the combination of lenvatinib–TACE is a promising treatment option for advanced HCC patients [49]. A phase III trial comparing lenvatinib and sorafenib for unresectable HCC involved 1492 patients; of those, 954 received lenvatinib (12 mg two times a day), and 538 received sorafenib (400 mg). Lenvatinib demonstrated a median OS of 13.6 months compared to the lenvatinib group and 12.3 months compared to the sorafenib group [50]. In this trial, patients with advanced HCC were randomly assigned to receive either lenvatinib–pembrolizumab (395 patients) or lenvatinib–placebo (399). Lenvatinib–pembrolizumab showed an mOS of 21.2 months compared to 19.0 months with lenvatinib alone. The combination therapy also demonstrated a higher ORR (26.1% vs. 17.5%) [51]. The combination therapy with lenvatinib has shown potential than sorafenib in the treatment of advanced HCC, as well as improved mOS and PFS, with a higher ORR than lenvatinib alone.

#### 2.1.3. Regorafenib

Regorafenib (MKI) was developed as the first choice of drug for HCC patients receiving sorafenib for advanced HCC as a second-line treatment [52]. Regorafenib generally acts as MKI, which interacts with angiogenic factors such as PDGFR-β, VEGFR1-3, and FGFR1, as well as oncogenes BRAF, cRAF/RAF-1, RET, and KIT [53]. Additionally, it alters the expression of proteins involved in the MAPK pathway (P-JNK, P-c-Jun, and P-ERK1/2), autophagy (Beclin-1 and LC3-II), and apoptosis (Bcl-2, Bax, Bcl-X, survivin, cleaved caspase-3, -7, -8, and -9) [54]. A more influential inhibitory activity is seen in regorafenib because it moderately blocks TIE2 and strongly inhibits c-KIT; therefore, it may be a more effective inhibitor of angiogenesis than sorafenib [55].

The results of a phase III trial on 573 patients previously treated with sorafenib who received regorafenib 160 mg/day compared to the placebo revealed that regorafenib was more effective than the placebo. The median time from sorafenib treatment until death was 26.0 months with regorafenib and 19.2 months with the placebo. This finding suggests that regorafenib provides high clinical outcomes regardless of prior sorafenib treatment with manageable AEs [56]. In a phase III trial on 843 HCC patients, after sorafenib treatment, the oral administration of regorafenib 160 mg once daily demonstrated improved OS compared to the placebo (10.6 months vs. 7.8 months). Regorafenib is a promising systemic treatment option for the HCC patient population, suggesting potential for combination therapies in future trials to optimize treatment outcomes beyond sorafenib and regorafenib alone [57]. A phase II trial was carried out in which regorafenib 160 mg was administered once daily in 40 unresectable HCC patients previously treated with atezolizumab–bevacizumab. The OS from the start of the atezolizumab–bevacizumab treatment was 16.6 months, and PFS was 3.8 months, which suggests that regorafenib could be a viable option for patients who received atezolizumab–bevacizumab therapy [58]. In a phase III trial, 43 patients with advanced HCC who progressed to sorafenib monotherapy were assigned to receive regorafenib 160 mg daily. The results demonstrated a significantly improved OS (17 months), and the median PFS was also longer (11 months). Regorafenib is well tolerated and effective in patients after sorafenib progression [59]. Regorafenib targets oncogenes and angiogenic factors, exhibiting an inhibitory activity against angiogenesis and tumor growth. A comprehensive overview of MKI and ICI drugs that have shown significant effects in HCC therapy in clinical trials with their specific molecular targets, modes of action, median OS, tumor-associated functions, and the specific inhibited pathways are listed in Table 1.

### 2.2. Tyrosine Kinase Inhibitors

#### Cabozantinib

Cabozantinib is also another TKI for MET and VEGFR2, but it also targets RET, KIT, and AXL. The biomarkers MET, ANG2, HGF, VEGF-A, GAS6, and IL-8 may be used to predict the prognosis of a patient after receiving cabozantinib [66]. Human cancer development has been linked to the tyrosine kinase receptor MET, which is encoded by the MET gene to interact with the hepatocyte growth factor (HGF) to activate downstream pathways and promote cell proliferation, migration, and angiogenesis [67]. Additionally, studies have evaluated the overexpression of MET in several human malignancies, including HCC. Gene therapy is used to inhibit MET expression, reducing proliferation, angiogenesis, and tumor development [68]. The blocking of MET and VEGFR pathways simultaneously may improve the effectiveness of HCC treatment [69]. 

Cabozantinib and the placebo were evaluated to assess the efficacy in a phase III trial. The cabozantinib group received 60 mg and showed an mOS of 10.2 months and a median PFS of 5.2 months in comparison to 8.0 months and 1.9 months of PFS in the placebo group [70]. A phase II trial evaluated the administration of cabozantinib 60 mg once daily in 48 HCC patients who progressed after ICI treatment. The mOS and PFS time were 9.9 months and 4.1 months with manageable AEs. Notably, patients who received prior atezolizumab–bevacizumab treatment had a longer mOS than those who did not [60]. In a phase II trial, 331 patients with advanced HCC and prior sorafenib therapy were randomized to receive cabozantinib 60 mg daily or a placebo. The cabozantinib group demonstrated 11.3 months of OS and an improved PFS compared to 7.2 months of OS in the placebo group. The results demonstrated the benefit of cabozantinib after sorafenib treatment in improving the survival rate of advanced-stage HCC patients [71]. These results highlight the potential of cabozantinib as a valuable therapeutic option for advanced-stage HCC patients, demonstrating its role in improving survival outcomes in this challenging HCC stage.

### 2.3. Angiogenesis Inhibitors

#### 2.3.1. Ramucirumab

Ramucirumab (anti-VEGF) is a monoclonal antibody-based recombinant IgG1 that specifically targets VEGFR2. Ramucirumab strongly binds to the human VEGFR-2 and blocks the ability of VEGFR-2 to bind to its ligands VEGF-A/C/D, which leads to the inhibition of cellular proliferation, migration, and permeability in HCC [72]. The high expression of VEGFR2 and VEGF is linked to the upregulation of alpha-fetoprotein (AFP) in HCC. Higher quantities of alpha-fetoprotein (400 ng/mL) have been linked to the overexpression of VEGFR and result in more angiogenesis in HCC. However, a high AFP level predicts a poor prognosis [73]. The FDA-approved ramucirumab was used as a second-line therapy for HCC patients in the first biomarker-based trial, which had positive outcomes in patients [16]. 

In a phase III trial, 565 advanced HCC patients who had previously received sorafenib were randomly assigned to receive ramucirumab 8 mg/kg or a placebo and their results were evaluated in terms of safety and efficacy. The mOS for ramucirumab was 9.2 months, whereas it was 7.6 months for the placebo [61]. A phase II trial involved 542 advanced HCC patients who received ramucirumab 8 mg/kg after two weeks or a placebo. Ramucirumab showed a survival benefit in patients with aggressive HCC and nonviral HCC [74]. Another phase III trial aimed to evaluate ramucirumab’s efficacy in advanced HCC patients and AFP concentrations of 400 ng/mL or higher. In total, 292 patients were randomized, and those receiving ramucirumab 8 mg/kg intravenously after 14 days demonstrated a significantly improved mOS (8.5 vs. 7.3 months) and PFS (2.8 vs. 1.6 months) and a manageable safety profile compared to the placebo group [75]. This study aimed to assess the efficacy of ramucirumab for unresectable HCC after lenvatinib failure. Ramucirumab has therapeutic potential for unresectable HCC in patients who have previously had lenvatinib therapy. The median PFS was improved, indicating a better therapeutic approach for HCC control [76]. These trials investigated the potential of ramucirumab as a promising treatment option for advanced-stage HCC, particularly in patients with prior sorafenib or lenvatinib therapy.

#### 2.3.2. Atezolizumab and Bevacizumab

Atezolizumab (anti-PD-L1) is a monoclonal IgG1 antibody and an inhibitor of programmed death ligand-1 (PD-L1) in cancer cells. It prevents PD-L1 binding to B7 and PD-1 on T cells. The blockage of PD-L1 binding increases the response of T cells [77]. Bevacizumab (anti-VEGF) is also a monoclonal antibody that alters the microenvironment of HCC through binding on the VEGF ligand to stop angiogenesis [78]. It prevents VEGF from binding to its receptor. Once the VEGF binds to a specific receptor, the defense mechanism of immune cells is suppressed, thus inhibiting the anticancer activity of the immune response [79]. Atezolizumab binds to the PD-L1 immunological checkpoint protein and is called PD-L1 inhibitor [80]. Combination therapy increases the survival rate; therefore, it is preferred over monotherapy [81]. Moreover, the occurrence of AEs was slightly higher in combination therapy than in monotherapy but not substantially different between the groups receiving monotherapy and combination therapy [82]. Therefore, atezolizumab with bevacizumab was proven to be safer for HCC patients. ICB therapy has been explored to switch off the immune checkpoint proteins for HCC treatment [83]. 

In a phase III trial, 336 patients with unresectable HCC were randomly assigned to receive atezolizumab–bevacizumab (1200 mg + 15 mg/kg once every 21 days). The results demonstrated a significantly improved OS (12 months), and the median PFS was also longer (6.8 months) in the atezolizumab–bevacizumab group [82]. The combined treatment of atezolizumab with bevacizumab is much more efficient than single sorafenib treatment in 501 patients of HCC who had not previously received systemic therapy. The combination therapy showed mOS of 19.2 months and a PFS of 6.9 months in contrast to 12.3 months with sorafenib; therefore, atezolizumab–bevacizumab continues to provide clinically significant survival advantages over sorafenib [64]. The treatment for advanced HCC has been reshaped by developing immunotherapy such as atezolizumab–bevacizumab with 90Y-TARE in the IMbrave150 trial. This combination is preferred for HCC patients with portal vein thrombosis but lacking distant metastases. Combining ICIs with VEGF antibody and 90Y-TARE has the potential for upregulating immune response [84]. The emergence of combination therapy using ICIs and anti-VEGF shows significant advancement in stimulating antitumor immune response and disrupting the tumor microenvironment, targeting both the immune checkpoint pathway and angiogenesis for HCC. Figure 3 provides a detailed overview of the primary molecular targets and signaling pathways involved in targeted therapy for HCC. The figure details the key functional molecules and their respective regulators within the signaling cascades that play crucial roles in HCC development.

### 2.4. Immune Checkpoint Inhibitors (ICIs)

PD-1 is a crucial immunological checkpoint and is widely expressed in myeloid, B, and T cells. The most significant advancement in cancer treatment during the past few years has been T-cell checkpoint inhibition [16]. Immune cells, i.e., macrophages, natural killer, T, and B cells, express membrane-bound molecules called immunological checkpoints to regulate the immune response [85]. Several checkpoint molecules, namely PD-1 on T cells, PD-L1 on tumor cells, and CTLA-4 on antigen-presenting cells or tumor cells, are most frequently researched in human cancer [86] (Figure 4). Chronic inflammation, intrinsic immunodeficiency, and high regulation of immune checkpoints such as the PD-1, PD-L1, and CTLA-4 pathways are all characteristics of the microenvironment where HCC frequently develops [87].

The PD-1/PD-L1 pathway is essential in suppressing tumor immunity and blocking this process by anti-PD-L1 antibodies. It might boost the antitumor immunity and slow down the tumor growth in several malignant tumors [88]. The interaction of PD-1 with PD-L1, sends a negative signal to T cells (against CD28), which can lead to immune suppression and tolerance to self-antigens [89]. The regulatory T cells (T_regs_) and myeloid-derived suppressor cells (MDSCs) are common immunosuppressive cells in the tumor environment. T_regs_ are a subpopulation of T cells that express TF Foxp3 (forkhead box p3) and the CD25 receptor of interleukin-2 (IL-2) [90]. The checkpoint molecules PD-1, PD-L1, CTLA-4, galectin-9, GITR (glucocorticoid-induced TNFR family-related gene), and Tim-3 (T-cell immunoglobulin and mucin-domain containing-3) interact with activated T_regs_ to inhibit various immune cells [91].

#### 2.4.1. Nivolumab

The prognosis is poor for HCC patients in whom PD-L1 is overexpressed. Nivolumab, a completely human anti-PD-1 antibody that binds to PD-1 on T cells to inhibit the interaction with PD-L1 and PD-L2, was tested in a phase I/II study in patients with HCC to determine its safety and preliminary anticancer efficacy [92]. The PD-1 ligand on tumor cells binds to the PD-1 receptor on T cells to inhibit T-cell proliferation and cytokine production. The blockage of PD-1 promotes responses from T cells and enhances antitumor immunity [86].

A phase III trial randomly assigned 743 patients into two groups: nivolumab vs. sorafenib. Nivolumab was equally effective as sorafenib to treat HCC. The mOS of nivolumab was improved to 16.4 months, and ORR was 15% compared to sorafenib, with an mOS of 14.7 months and 7% ORR. Nivolumab was also safer with fewer AEs than sorafenib [93]. In a phase III trial, patients undergoing neoadjuvant therapy before LR in early-stage HCC were randomly assigned to receive nivolumab and ipilimumab. Nivolumab 1 mg/kg was administered for a duration of two 21-day cycles (i.e., six weeks of therapy) on days 1 and 22 and ipilimumab 3 mg/kg only on the first day of a 21-day cycle. The results demonstrated a significantly improved OS and PFS [94]. A phase II trial aimed to evaluate nivolumab efficacy as adjuvant therapy after RFA and surgery on 55 intermediate-stage HCC patients. Nivolumab (480 mg once every 1 month after RFA or surgery) demonstrated significantly improved recurrence-free survival (RFS) (26 months); therefore, this study likely precedes further extensive or long-term trials that would define this drug as a late-stage treatment option [95]. 

#### 2.4.2. Pembrolizumab

Pembrolizumab is a strong and specific IgG4 monoclonal antibody used to prevent the direct binding between PD-1 and its ligands PD-L1 and PD-L2. It blocks the PD-1 receptor on T cells, making the immune system attack cancer cells more effectively [96]. Pembrolizumab exhibited a positive response and controllable toxicity with low AEs in HCC patients [97]. In a phase I trial on 32 intermediate-stage HCC patients receiving TACE–pembrolizumab (200 mg), combining TACE with pembrolizumab showed a manageable safety profile with no synergistic toxicity. Some patients had stable disease on pembrolizumab, suggesting potential efficacy in advanced HCC patients [98].

A phase II trial on 104 advanced HCC patients enrolled at 47 locations across 10 countries divided them into the pembrolizumab group and the placebo group. The placebo group showed less OS and PFS than the pembrolizumab group (OS 10.6 months and PFS 2.8 months in the placebo group vs. OS 13.9 months and PFS 3 months in the pembrolizumab group). The ORR in the pembrolizumab group was 18.3% and considerably greater than the 4.4% ORR of the placebo group. The results of the first phase III trial support the approval of pembrolizumab and demonstrate the effectiveness of anti-PD-1 antibody in the treatment of HCC [99]. Another phase III trial evaluated the efficacy of pembrolizumab in 453 patients with advanced-stage HCC. Patients were randomly assigned to receive pembrolizumab or a placebo. Pembrolizumab demonstrated improved mOS (14.6 vs. 13.0 months) and median PFS (2.6 vs. 2.3 months) compared to the placebo. Additionally, the ORR was higher in the pembrolizumab group (12.7% vs. 1.3%). These findings suggest that pembrolizumab is a valuable treatment option for previously treated advanced HCC patients [63]. A phase III trial randomized 750 advanced HCC patients in two groups: one receiving lenvatinib (12 mg daily)–pembrolizumab (200 mg) and another receiving lenvatinib–placebo. The safety and efficacy of lenvatinib–pembrolizumab were higher than those of lenvatinib–placebo as first-line therapy for advanced HCC [100]. These findings suggest that pembrolizumab is a valuable treatment option for advanced HCC patients, making the path for further clinical trials to manage this challenging HCC stage. Figure 5 illustrates the mechanisms of action for different ICIs targeting their specific immune checkpoint proteins.

#### 2.4.3. Nivolumab and Ipilimumab

Combination therapy using ICIs is a successful HCC therapy method. PD-1 and CTLA-4 are inhibitory immune checkpoints present on activated T cells and have been proposed as targets for HCC therapy [101]. Several ICIs such as pembrolizumab and nivolumab target PD-1, while atezolizumab and durvalumab target PD-L1. Ipilimumab and tremelimumab were approved for targeting CTLA-4 in several cancers [102]. ICI monotherapy was less effective in HCC patients. The combined use of nivolumab and ipilimumab is often researched to improve oncological outcomes to combat HCC [103]. Nivolumab and ipilimumab were approved in a phase II clinical trial that randomly assigned patients with unresectable or metastatic HCC who received sorafenib treatment in the past or were intolerant to it to receive nivolumab and ipilimumab. The results of this trial revealed that the ORR was 32% and mOS was 22.8 months with a dosage regimen comprising nivolumab 1 mg/kg + ipilimumab 3 mg/kg every 21 days followed by nivolumab 240 mg once every 14 days. Nivolumab in combination with ipilimumab demonstrated a promising ORR and long-lasting effects [104]. 

A phase I trial investigated nivolumab–ipilimumab combination as a neoadjuvant therapy before LR in early-stage HCC and showed clinically significant results. The ORR (31%) and OS of the combination therapy for HCC had been improved by combination immunotherapy [94]. This randomized phase II trial investigated the efficacy of neoadjuvant immunotherapy in 30 patients with resectable HCC. Patients were assigned to receive either nivolumab alone (240 mg every 14 days) or nivolumab–ipilimumab (140 mg + 1 mg/kg every 1.5 months) before surgery. Pathologic complete response was observed in 24% of patients, with an additional 16% achieving a major pathologic response. These findings suggest that neoadjuvant immunotherapy holds promise for resectable HCC and requires further trials [105]. A CheckMate 040 clinical trial evaluated the use of combination immunotherapy by nivolumab (1 mg/kg)–ipilimumab (3 mg/kg) on 148 advanced HCC patients. A median OS of 22.2 months, an ORR of 32%, and a high disease control rate with manageable AEs were recorded. Patients with HBV or HCV-related HCC had higher ORR than uninfected HCC patients. This trial showed significant responses and long-term survival benefits in patients with advanced HCC [62]. Trials investigating the combination of nivolumab and ipilimumab have demonstrated promising ORR and prolonged OS in advanced or unresectable HCC patients.

#### 2.4.4. Durvalumab and Tremelimumab

Durvalumab and tremelimumab were successful as anti-PD-L1 and anti-CTLA-4 combined immunotherapy for HCC [106]. Durvalumab and tremelimumab activate naive T cells in T-cell stimulation, increasing the number of activated cytotoxic T cells in the blood [107]. Tremelimumab increases the immunosuppression cancer microenvironment by inhibiting Treg and MDSCs to promote the death of cancer cells [108]. The combined anti-CTLA-4 and anti-PD-L1 antibodies activate and increase CD8^+^ T cells to enter the tumor and decrease Tregs and MDSCs [109].

The use of combined ICIs has demonstrated significant promise in treating advanced HCC. However, there is a need to explore more than two checkpoint inhibitors (CPIs) while managing treatment-related toxicity. A phase II trial evaluated this by assessing the efficacy of triple treatment with a combination of durvalumab, tremelimumab, and bevacizumab in patients with advanced HCC. The results demonstrated the ORR and OS were high in the triple combination. This approach represents a novel strategy for optimizing immunotherapy in HCC treatment [110]. Another phase III trial evaluated durvalumab–tremelimumab efficacy in 300 patients with unresectable HCC. Sorafenib-treated patients with previously untreated locally progressed or metastatic HCC were administered tremelimumab 300 mg and durvalumab 1500 mg every 1 month [65]. A clinical trial investigated the safety of combining tremelimumab (75 mg every 1 month for 4 doses)–durvalumab (1500 mg every 28 days) and TACE in 13 patients with advanced HCC. Tremelimumab–durvalumab with TACE showed safety in HCC patients [111]. The results of these studies are not available yet as these trials are currently in the screening phase. The combination therapy showed a significant advancement in immunotherapy for HCC treatment. However, further research is ongoing to fully understand the efficacy and safety profile of double and triple ICI combinations (Figure 6).

## 3. Conclusions

Hepatocellular carcinoma continues to be one of the deadliest cancers with a low rate of survival; therefore, currently, a very active area of study is the treatment of advanced HCC. The current understanding of genetic pathways and the analysis of gene expression has provided a systematic mechanism to understand central metabolic, signaling, and carcinogenic pathways in HCC. This has led to the development of specific targets to control HCC. The discovery of TKIs followed by immunomodulators has increased the life expectancy and OS rate of HCC patients. Drugs targeting PD-1, PD-L1, and CTLA-4 have shown promising results in clinical trials, creating new hope for HCC patients. Regorafenib, nivolumab, ramucirumab, and cabozantinib are now accessible and effective choices in monotherapy. The combined therapy of TKI–ICI, ICI–ICI, and ICI–anti-VEGF therapy has dramatically altered the course of HCC with a high survival rate. The combined use of atezolizumab–bevacizumab, nivolumab–ipilimumab, and durvalumab–tremelimumab changes the current scenario for HCC treatment. Several studies reported concerns about a rise in the recurrence of HCC and explored new avenues of drugs in combination therapy. An effective antitumor activity and toxicity must be balanced during the course of the treatment with novel therapeutic substances and therapies. It is still unknown which HCC patients respond best to a particular drug and what the best therapy order is. This review highlights FDA-approved drugs in monotherapy and combination therapy for targeting molecular pathways and immune checkpoints for possible HCC treatment. Some ongoing research aims to evaluate the mechanisms underlying HCC recurrence and resistance to therapy and explore combination (double and triple) therapies and immunotherapeutic approaches to enhance OS, PFS, and ORR with fewer AEs.

## Figures and Tables

**Figure 1 cancers-16-02034-f001:**
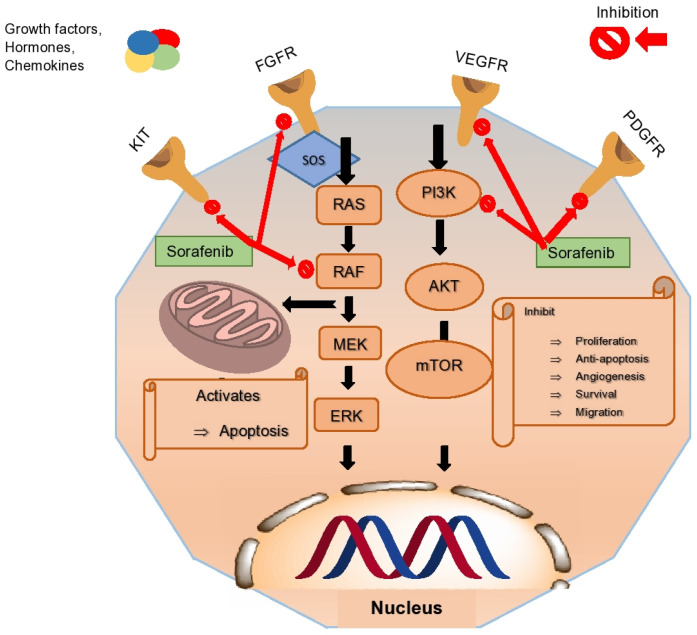
Effect of sorafenib on the RAS/RAF cascade and specific receptors to initiate apoptosis. The drug inhibits phosphorylation of eIF4E to stop the initiation of transcription of RAS/RAF cascades. The resultant deficiency of proteins prevents cell division to stop tumor growth.

**Figure 2 cancers-16-02034-f002:**
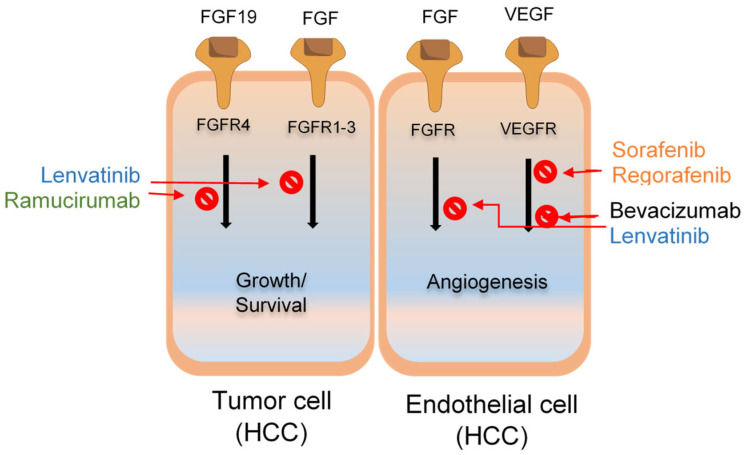
Lenvatinib inhibits both the vascular endothelial growth factor and fibroblast growth factor pathways to stop the development, survival, the proliferation of tumor cells, and endothelial cells involved in angiogenesis.

**Figure 3 cancers-16-02034-f003:**
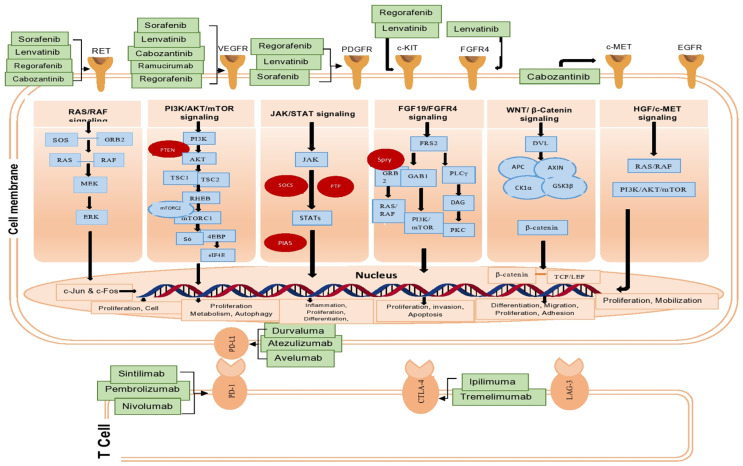
The primary molecular targets and signaling pathways for HCC targeted therapy. We listed the functional molecules with their regulators in signaling cascades as well as significant signaling pathways that are important for HCC development. Finally, we listed the targeted drugs currently being tested in clinical trials for advanced-stage HCC. SOS, son of sevenless; GRB2, growth factor receptor-bound protein 2; eIF4E, eukaryotic translation initiation factor 4E; RHEB, RAS homolog enriched in brain; PTEN, phosphatase and tensin homolog; TSC1/2, tuberous sclerosis ½; S6K, S6 kinase; 4EBP1/2, eukaryotic translation initiation factor 4E-binding protein; mTORC1/2, mammalian target of rapamycin complex ½; ½ JAKs, Janus kinase; SOCS, suppressors of cytokine signaling; STAT, signal transducers and activators of transcription; PIAS, protein inhibitors of activated STAT; PTP, protein tyrosine phosphatase; FGFR4, fibroblast growth factor receptor-4; FGF19, fibroblast growth factor-19; FRS2/3, fibroblast growth factor receptor substrate 2 and 3; GAB1, GRB2-associated binding protein 1; DAG, diacylglycerol; PLCγ, phospholipase C gamma; c-MET, mesenchymal–epithelial transition factor; WNT, wingless related integration site; CK1α, casein kinase 1α; DVL, disheveled; APC, adenomatous polyposis coli; GSK3β, glycogen synthase kinase 3β.

**Figure 4 cancers-16-02034-f004:**
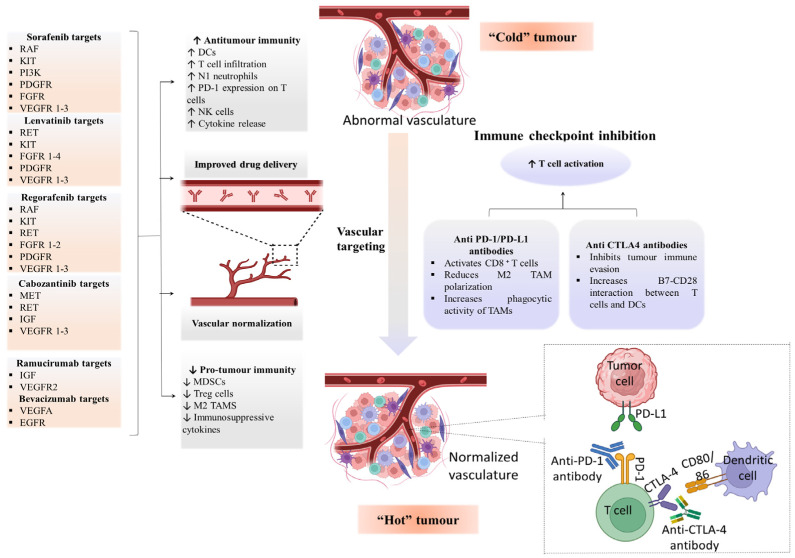
The synergistic therapeutic efficacy of antiangiogenic agents and immune checkpoint inhibitors involves unraveling underlying mechanistic insights. Combining immune checkpoint inhibitors with tyrosine kinase inhibitors or antiangiogenic antibodies together with locoregional treatment might increase response rates by infiltrating immune cells into “cold” tumors and transforming them into “hot” tumors. This synergy might involve several processes, including the activation of different anti-immune cells, the inhibition of immune cell types that promote tumor growth, or vascular normalization, and enhances immunological infiltration and drug delivery.

**Figure 5 cancers-16-02034-f005:**
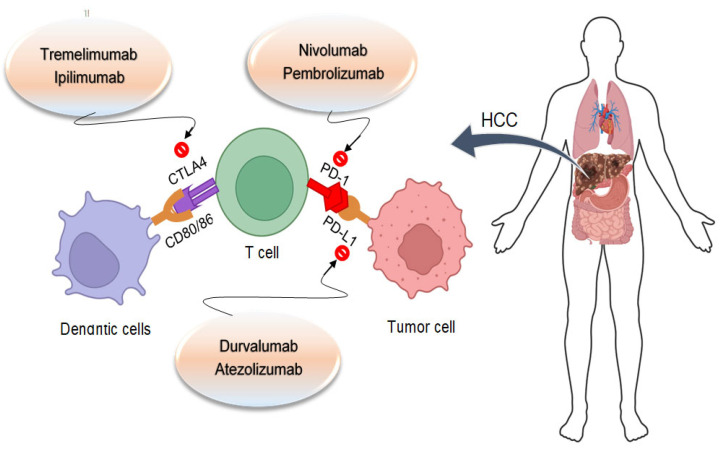
PD-1 binds with PDL1 to inhibit the growth and development of T-killer cells and causes T cells to become exhausted. CTLA4 attaches to CD80/86 to prevent T cells from becoming activated. Immunological checkpoint inhibition prevents immunity exhaustion, diminishes regulatory T cell activity, and triggers the recurrence of the antitumor immunological response.

**Figure 6 cancers-16-02034-f006:**
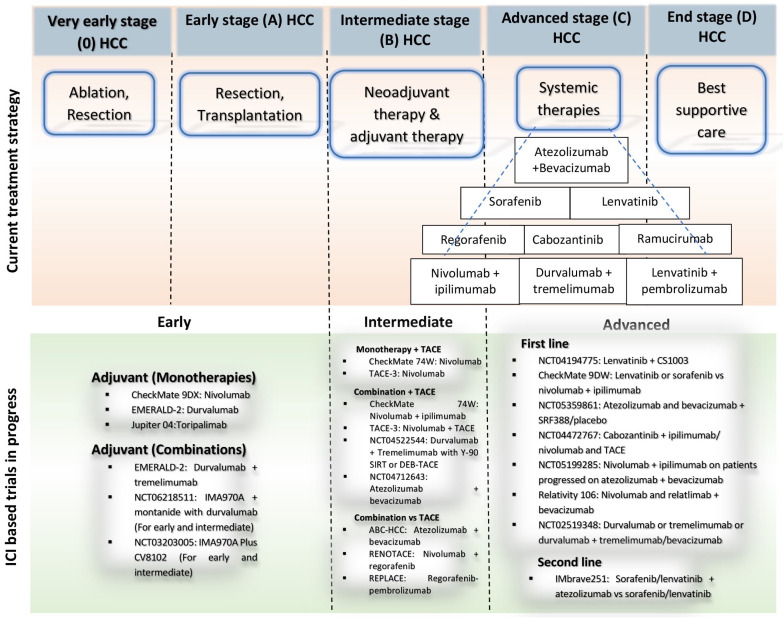
Natural history, current treatment approaches, and future advancements in immunotherapy for the effective management of HCC. The Barcelona Clinic Liver Cancer (BCLC) staging system categorizes HCC into five stages based on factors such as the extent of disease and liver function. These are some ongoing trials for early-stage HCC investigating the safety and efficacy of drugs in monotherapy or combination therapy for recurrence-free survival. In intermediate and advanced stages of HCC, ICIs are often used in combination with targeted therapy to increase the overall response and objective response rate.

**Table 1 cancers-16-02034-t001:** Drugs shown in clinical trials to have a significant effect on HCC therapy.

Drugs	Commercial Name	Clinical ID	Phase	Targets	Mode of Action	Medium Overall Survival (Months)	Tumor-Associated Function	Specific Inhibited Pathway	Ref.
**Sorafenib**	Nexavar	NCT01015833	III	RAF, VEGFR, PDGFR-β, TGF, IGF, FLT-3	MKI	mOS 9.4 months	Immunoregulation, proliferation, angiogenesis	KIT, RET, FGFR1, VEGFR2/3, PDGFR-β	[40]
**Lenvatinib**	Lenvatinibvima	NCT01761266	III	KIT, RET, PD-1, TGF, IGF, FGFR1-4, VEGFR1-3, PDGFR-α	MKI	mOS 13.6 months	Immunoregulation, proliferation, angiogenesis	KIT, RET, FGFR1–4, VEGFR1–3, PDGFR-α	[50]
**Regorafenib**	Stivarga	NCT01774344	III	B-RAF, KIT, RET, TGF, IGF, FGFR1, VEGFR1-3, PDGFR-β	MKI	mOS 10.6 months	Immunoregulation, proliferation, angiogenesis	KIT, RET, VEGFR1–3, PDGFR-β	[57]
**Cabozantinib**	Cabozantinibmetyx and Cometriq	NCT04588051	II	AXL, MET, TGF, IGF, VEGFR2	TKI	mOS 9.9 months	Metastatic lesions, migration, angiogenesis	KIT, RET, AXL, MET, VEGFR2	[60]
**Ramucirumab**	Cyramza	NCT01140347	III	TGF, IGF, VEGFR2	mAb	mOS 9.2 months	Immunoregulation, angiogenesis	VEGFR2	[61]
**Nivolumab + ipilimumab**	Opdivo + Yervoy	NCT01658878	II	PD-1 + CTLA-4	Anti-PD-1 antibody + Anti-CTLA-4-antibody	mOS 22.2 months	Proliferation, angiogenesis, immunoregulation	PD-1, CTLA-4 pathway	[62]
**Pembrolizumab**	Keytruda	NCT03062358	III	PD-1	Anti-PD-1 anibody	mOS 14.6 months	Tumor regression, immune regulation	PD-1	[63]
**Atezolizumab + bevacizumab**	Tecentriq + Avastin	NCT03434379	III	PD-L1 + VEGF	Anti-PD-L1 antibody + Anti-VEGF-antibody	mOS of atezolizumab + bevacizumaba 19.2 months	Proliferation, metastasis, invasion, angiogenesis	PD-L1, VEGF	[64]
**Durvalumab + tremelimumab**	Imfinzi + Imjudo	CTR20222433	III	PD-L1 + CTLA-4	Anti-PD-L1 antibody + Anti-CTLA-4-antibody	N/A	Immunoregulation, proliferation, angiogenesis, necrosis	PD-L1, CTLA-4	[65]

## Data Availability

Not applicable.

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
