# Peer review of "Hepatocellular Carcinoma: Beyond the Border of Advanced Stage Therapy"

_cancers, 2024, doi:10.3390/cancers16112034_

Round 1

Reviewer 1 Report

Comments and Suggestions for Authors

Interesting paper on a cutting-edge topic. The paper deals with systemic therapies for HCC so the title should reflect the content of the manuscript. Also, i would remove the first paragraph that it is only a short overview on ablative treatments and it seems out of context.

The authors should comment more on the comparative efficacy of lenvatinib over sorafenib, citing the recent SRMA in the field (PMID: 34017396)

A table with the ongoing trials on these drugs would be helpful.

Figure 2 reports the mechanism of action of only few molecules.......you should add also the other drugs in the figures or to create another image

Author Response

Thank you for taking the time to review our manuscript titled "Hepatocellular Carcinoma: Beyond the Border of Advanced Stage Therapy" and for providing valuable feedback. We appreciate your constructive comments, which have greatly contributed to improving the quality of our work. Below in word file, we address each of your points in detail.

Thank you for the time and consideration

Regards,

Abdul Ghaffar

Reviewer 2 Report

Comments and Suggestions for Authors

Zarlashat et al. reviewed the use of TKI and ICIs in the treatment of patients with HCC.

This is well a paper a well written and easy to read however some points need to be clarified

1) please use the new nomenclature for MAFLD and NASH

2) it would be useful to expand the part regarding the use of systemic therapy in the adjuvant setting also by adding a table on the ongoing trials to date (please consider the data from conferences not yet published as full papers). For example the EMERALD-1 study of durvalumab plus bevacizumab plus TACE should be included

3) it would be useful to discuss the concept of personalized therapy and the "multiparametric therapeutic hierarchy" in the treatment of HCC (ref Vitale et al Lancet Oncology)

4) It is useful to include also data from the real practice on the use of TKIs

5) please include a specific chapter on the safety and tolerability of these treatments

Author Response

Thank you for giving your time to review our manuscript titled "Hepatocellular Carcinoma: Beyond the Border of Advanced Stage Therapy" and for providing comments. We appreciate it, which has contributed to improving the quality of our work. Below in word file, we address each of your points in detail. We have carefully considered all of your suggestions and have made the revisions.

Once again, we sincerely appreciate your time and effort in reviewing our work. If you have any further questions or require additional information, please don't hesitate to contact us.

Regards,

Abdul Ghaffar

Reviewer 3 Report

Comments and Suggestions for Authors

The manuscript entitled " Hepatocellular Carcinoma: Beyond the Border of Therapy " was reviewed.

The description in the Introduction is general; there is no doubt that new agents are expected for advanced stage Hepatocellular Carcinoma. Clinicians have experienced the limitations of drug-only therapy for advanced stage Hepatocellular Carcinoma. In other words, we know that it is not very promising.On the other hand, Hepatocellular Carcinoma is treated with TACE, which is not performed in other carcinomas. As the authors also describe in Figure 5, the combination of TACE and drug therapy is also being actively used. Why did the authors focus this paper on drugs? Are there any new treatments or new trials that the authors are conducting for advanced stage Hepatocellular Carcinoma?

Could Radiation segmentectomy (Cancers 2024, 16(3), 669) be added to the Emerging Therapeutic Options for HCC Therapy section of this paper?

Comments on the Quality of English Language

none

Author Response

Dear Reviewer,

Thank you for the time to review our manuscript titled "Hepatocellular Carcinoma: beyond the Border of Advanced Stage Therapy." We appreciate your thoughtful feedback and constructive suggestions. We have carefully considered all of your suggestions and have made the revisions which are listed in the below word file.

Thank you once again for your valuable input. Should you have any further questions or require additional clarification, please don't hesitate to contact us.

Regards,

Abdul Ghaffar

Round 2

Reviewer 1 Report

Comments and Suggestions for Authors

The revised version of the manuscript is OK. Thank you!

Reviewer 2 Report

Comments and Suggestions for Authors

Authors answered to all my questions and the paper significantly improved